# Inventory of Patient-Reported Outcome Measures Used in the Non-Operative Care of Scoliosis: A Scoping Review

**DOI:** 10.3390/children10020239

**Published:** 2023-01-29

**Authors:** Eric C. Parent, Matthew Vaclavik, Cody Bourgoin, Courtney Hebert, Megan Bouwmeester, Sarah Cheslock, Rebecca Collins, Stefan Potgieter, Mark Coles, Sanja Schreiber, Sabrina Donzelli, Camille Warner

**Affiliations:** 1Department of Physical Therapy, Faculty of Rehabilitation Medicine, University of Alberta, 2-50 Corbett Hall, Edmonton, AB T6G 2G4, Canada; 2The Foundation for Student Science and Technology, Ottawa, ON K1P 5J3, Canada; 3Italian Scientific Spine Institute (ISICO), Via Bellarmino 13/1, 20141 Milano, Italy

**Keywords:** scoliosis, conservative, non-operative treatment, patient-reported outcome measures, review, quality of life, physical appearance, function, pain, language

## Abstract

It is unclear which patient-reported outcome measures (PROMs) can assess non-operative care for scoliosis. Most existing tools aim to assess the effects of surgery. This scoping review aimed to inventory the PROMs used to assess non-operative scoliosis treatment by population and languages. We searched Medline (OVID) as per COSMIN guidelines. Studies were included if patients were diagnosed with idiopathic scoliosis or adult degenerative scoliosis and used PROMs. Studies without quantitative data or reporting on fewer than 10 participants were excluded. Nine reviewers extracted the PROMs used, the population(s), language(s), and study setting(s). We screened 3724 titles and abstracts. Of these, the full texts of 900 articles were assessed. Data were extracted from 488 studies, in which 145 PROMs were identified across 22 languages and 5 populations (Adolescent Idiopathic Scoliosis, Adult Degenerative Scoliosis, Adult Idiopathic Scoliosis, Adult Spine Deformity, and an Unclear category). Overall, the most used PROMs were the Oswestry Disability Index (ODI, 37.3%), Scoliosis Research Society-22 (SRS-22, 34.8%), and the Short Form-36 (SF-36, 20.1%), but the frequency varied by population. It is now necessary to determine the PROMs that demonstrate the best measurement properties in the non-operative treatment of scoliosis to include in a core set of outcomes.

## 1. Introduction

Adolescent idiopathic scoliosis (AIS) is a three-dimensional structural spine disorder that consists of rotation of the vertebrae presenting with a lateral spine curvature ≥10 degrees [1]. It affects 0.47% to 5.2% of adolescents worldwide and is more prevalent and severe in females [2]. Depending on the magnitude of the curve, the choice and cost of treatment can vary. For some patients, exercise alone or with a brace may be the only treatment required, but for others, expensive and invasive surgery may be necessary.

Adult spinal deformity (ASD) is still commonly used to describe a variety of conditions, including two types of adult scoliosis: idiopathic and degenerative (de novo) [3]. Both degenerative and adult idiopathic scoliosis can cause back pain [3]. Severe curves (over 80 degrees) of any type may impair pulmonary function [4]. Adult idiopathic scoliosis is a continuation in adulthood of AIS, which often presents as a right thoracic with left thoracolumbar or lumbar curve [3]. Prevalence of scoliosis in adults over the age of 25 is approximately 8.3% [5]. Degenerative (de novo) scoliosis is typically seen in patients over the age of 60 and mainly manifests as a lumbar curve [6]. Degeneration in the intervertebral discs and facet joints causes the degenerative scoliotic curvature. The angle of curvature may increase as patients age and their vertebrae further deteriorate [6]. In a study of 75 elderly participants with no previously known structural spine changes, radiographic evidence of scoliosis was detected in 68% of participants. It was suggested that deterioration of the spine caused degenerative scoliosis [7].

Individuals with scoliosis face health challenges. A Japanese study of 43,000 adolescents showed that those with AIS were two times more likely to experience back pain when compared to those without AIS [8]. The intensity of the pain experienced also increased in patients with scoliosis, with 24.2% rating their pain as severe enough to stay home from school [8]. In contrast, only 3.8% of the 10,561 students without scoliosis ranked their pain at the same level [8].

Lower self-esteem and decreased perception of one’s physical appearance, as well as reduced mobility, may result from scoliosis. Depending on the severity of the spinal curvature, surgery may be required to prevent the curve from worsening. A study examined 156 patients who had undergone surgery and 127 patients treated with a brace, finding that 49% and 34% of the surgery and brace groups, respectively, experienced difficulty participating in social activities [9]. This occurred due to self-consciousness about their appearance or reduced physical functioning because of their curve. By contrast, only 15% of controls experienced difficulties participating in social activities [9]. 

It is, therefore, relevant to analyze the use of various patient- and proxy-reported outcome measures (PROMs) for scoliosis. PROMs often take the form of questionnaires to assess the patient’s health status [10]. Deciding which PROM to use to best assess health-related quality of life (HRQOL) or other relevant outcomes is a difficult task. Many tools are available, and it is unclear which produce the best measurements. Additionally, language translation is needed for many PROMs and is, therefore, a barrier to gathering global data in patients with scoliosis. 

Many PROMs, such as the Scoliosis Research Society-22 (SRS-22) and Scoliosis Quality of Life Index, have been developed for patients undergoing surgery [11,12]. They present adequate measurement properties for assessment of those with severe curves but are limited in capturing the impact of smaller curves [13,14]. From these questionnaires, patients with smaller curves, especially adolescents, may appear to have no impact from scoliosis because these PROMs exhibit ceiling effects on multiple domains [13,14,15]. Ceiling effects also limit the ability of the PROMS to identify changes in patients with a small curve magnitude, as the patients do not have room to demonstrate improvement on such scales.

To ensure comparison of results, worldwide adoption of core outcome sets is necessary to assess treatments for scoliosis. The Nordic Spinal Deformity Society (NSDS) proposed 13 core outcome domains [16]. However, the core outcome domains targeted patients who had undergone surgery. The study excluded any data related to non-operative care. Furthermore, only seven spine surgeons representing only the Nordic countries voted in the Delphi study. From the core set, only 10 of the selected domains could be used to assess non-operative care. Developing Core Outcome sets has become a topic of interest worldwide, and at present, Close et al. are conducting a qualitative study to develop a person-centered core set for adolescents and young adults with spinal deformity undergoing treatment [17].

In a recent systematic review, the types of questions in PROMs used for patients undergoing surgery for ASD were analyzed and matched to 29 of the International Classification of Functioning and Health (ICF) measurement domains [18]. Of the core outcome sets proposed to date, none are targeted at non-operative care. To ensure that a core outcome set applies to all patients with AIS and adult spinal alignment disorders, it is imperative that patients receiving non-operative care are considered in future core outcome set proposals.

A scoping review that inventories all available PROMs with the available translations and that highlights those that are most popular will provide a basis for a systematic review of the measurement quality of such measurement tools. This will help clinicians identify relevant tools to monitor non-operative care in different languages to document the effects of non-operative treatments for AIS and adult spinal alignment disorders. This may ultimately help inform a core outcome set recommendation with adequate measurement properties, which could be used for a collaborative international database. 

The objective of this scoping review was to create an inventory of the PROM tools used in monitoring the effects of non-operative care management of AIS and adults with structural spine disorders. We aimed to list the types of PROMs used based on the population, language and to document the stages of care in which the PROMs were most often administered. 

## 2. Materials and Methods

### 2.1. Databases

A MEDLINE search strategy (Appendix B: Table A1. Medline Search Strategy) was implemented through OVID. Medline was searched from its inception (1946) to 15 January 2020. 

### 2.2. Search Strategy

Our search was informed by reviewing the search strategies of review protocols on PROSPERO, published systematic reviews of the measurements in similar populations, and proposed search filters to identify literature on PROMs [19]. Finally, we collected input from reviewers, including the perspectives of scoliosis experts in nursing, medicine, physical therapy, kinesiology, and library sciences. The search followed the recommendation by the COnsensus-based Standards for the selection of health Measurement INstruments (COSMIN) for searches aiming to investigate all PROMs for a population [20]. We identified terms related to scoliosis, the population, and PROMs. To limit the yield to the most relevant references, we eliminated populations in which scoliosis was a symptom of another disease by using the terms syndrome, tetraplegi*, and paraplegi*. We also excluded terms for common neuromuscular or congenital diseases as listed in Appendix B (Table A1. Medline Search Strategy). 

References found were collated in Covidence, and exact duplicates were eliminated. Close duplicates were inspected by one reviewer, and any additional duplicates were removed.

### 2.3. Screening Process

Selection criteria were developed by the review team based on recommendations from the COSMIN manual for systematic reviews of PROMs [20] (Table 1).

After eliminating duplicates, references from the search were uploaded into Covidence systematic review software (Veritas Health Innovation, Melbourne, Australia; available at https://www.covidence.org, accessed on 24 January 2023) and assigned to a team of reviewers for screening. If an article met the selection criteria or the reviewer could not ascertain if it met the criteria based on the abstract, they would include the article in the full-text review stage. The PDFs of the articles selected for full-text review were then uploaded into Covidence. The full texts were divided among the team of reviewers such that each reviewer was given a target of 100 articles. For this scoping review, only 1 reviewer was asked to screen references at both the title/abstract and the full-text screening stages. 

### 2.4. Data Extraction

For articles meeting the selection criteria, one of the reviewers extracted population data, the names of any PROMs used, and the languages and stage of care (observation, exercise, bracing, unspecified non-operative or pre-operative) in which the PROMs were administered. These data were collected using a shared online spreadsheet. Where relevant, extraction was summarized for each of the following four population groups as identified in the included studies and combining all these scoliosis populations: AIS, adult degenerative scoliosis (ADS), adult with idiopathic scoliosis (Adult IS), Adult spinal deformity (ASD), and Unclear. The Unclear category included patients with scoliosis who did not meet exclusion criteria but for whom we could not confidently assign a specific diagnosis category based on the published sample description.

### 2.5. Data Synthesis

A list of all the PROMs used with the number of studies in which they were used was compiled for all patient groups, as well as for each of the populations, languages, and stages of care in which the PROMs were used (Appendix A).

## 3. Results

### 3.1. Article Search and Selection

Our search in Medline found 3738 articles (Figure 1). After discarding 14 duplicates, 3724 studies were examined during title and abstract screening; 2824 were deemed irrelevant; and 900 were included for full-text screening. After full-text screening, 488 were included and underwent data extraction, whereas 412 studies were excluded for the following: containing other diagnoses in more than 20% of the sample than purely AIS, Adult IS, or ADS (*n* = 134); not including quantitative data (117); not including PROMS of relevance to non-operative care (54); incorrect administration of PROM (41, e.g., peri or post-op only); irrelevant study design (32 designs listed under exclusion criteria); and other less common reasons (Figure 1). Only 15 references were excluded because a full-text version of the article could not be retrieved at our library, via inter-library loan, or by attempting to contact the authors.

### 3.2. PROM Inventory

#### 3.2.1. Number and List of PROMs Inventoried

A total of 145 patient-reported outcome measure (PROM) tools were identified in 22 different languages over 5 patient populations (Table 2). In total, 120 PROMs were used for AIS, 26 for ADS, 20 for Adult IS, and 31 for ASD, and 33 PROMs were used in the Unclear category.

#### 3.2.2. PROMs Inventoried Combining All Populations

Overall, combining publications on all the populations inventoried, the most frequently used PROMs were (Table 3): the Oswestry Disability Index (ODI) (37.3% of studies), SRS-22 (34.8%), Short Form—36 (SF-36) (20.1%), Visual Analogue Scale for pain (VAS) (15.4%), SRS-22 revised (SRS-22r) (14.3%), SRS-24 (5.5%), Spinal Appearance Questionnaire (SAQ) (5.5%), Bad Sobernheim Stress Questionnaire-Deformity (BSSQ) (4.9%), SRS-30 (4.3%), and the Short Form-12 (SF-12) (4.3%). Various versions of the SRS questionnaire have been used in 58.9% of the studies.

#### 3.2.3. PROMs Inventoried Separately for Each Population

The most frequently used PROMs when divided by population also appear in Table 2. When authors did not specifically identify which type of scoliosis was included in the sample, the most frequently used PROMs were the ODI, followed by the SRS-22 and then the VAS.

Some PROMS used in only one study do not appear in Table 2 and are listed here grouped by the population in which they were used. In the AIS population, the following tools were used in only one study each: 0–10 Modified Borg Scale of the degree of dyspnea; 18-item Bracing-Beliefs Questionnaire (BBQ); Activities of Daily Living (ADL); Activity Performance; Adolescent Health Survey; Anger Expression Scale (AEX); Back-pain-specific version of the Hannover Functional Ability Questionnaire (HFAQ); Battle Culture-free Self-esteem Index for Children and Adolescents; Berner Questionnaire for Well-Being (BFW); Body Cathexis Scale; Body Esteem Scale for Adolescents and Adults (BESAA) Questionnaire; Bracing-Related Questions; Brief Pain Inventory Questionnaire (BPI); Center for Epidemiological Studies Depression Scale for Children (CES-DC); Climent Quality of Life for Spinal Deformities Scale; Comorbid Somatic Symptom Severity (SSS); Copenhagen Neck Functional Disability Scale (CNFDS); Depression, Anxiety, and Stress Scale-21 (DASS-21); Disabilities of the Arm, Shoulder, and Hand (DASH); Douleur Neuropathique 4 (DN4); Epworth Sleepiness Scale (ESS); Erich Mittenecker and Walter Toman Personality Test; Female Sexual Function Index (FSFI); Fibromyalgia Survey Criteria; Functional disability inventory (FDI); Functional Impairment; Functional Independence Measure for Children scale (WeeFIM); Generalized Anxiety Disorder 7-item scale (GAD-7); Generic KIDSCREEN-52; Global Perceived Effect (GPE); High School Personality Questionnaire (HSPQ); Insomnia Severity Index (ISI); International Physical Activity Questionnaire Short Form (IPAQ-SF); Italian Spine Youth Quality of Life (ISYQOL); Kidcope; Modified Life satisfaction index Z scale; Modified Scoliosis Research Society Outcome Instrument (MSRSI); Mood scale (0-10); Multidimensional Body-Self Relations Questionnaire (MBSRQ); Multisite Pain; Odense Scoliosis Questionnaire (OUH); Offer Self-Image Questionnaire Revised (OFFER); Pain body diagram; Pain Control Beliefs Questionnaire; Pain drawing; Pain questionnaire (peak, general level, frequency); PainDETECT; Pediatric Evaluation of Disability Inventory (PEDI) fixed forms; Pediatric Anxiety Rating Scale (PARS); PEDI-Multidimensional Computerized Adaptive Testing (MCAT); Perception of Back by Drawing Test; Perseverative Thinking Questionnaire (PTQ); Pittsburgh Sleep Quality Index (PSQI); Prevalidated questionnaire; Psychological General Well-Being (PGWB); Quebec Back Pain Disability Scale (QDS); Revised Child Anxiety and Depression Scale; Revised Oswestry Disability Index (RODI); Rosenberg Self-Esteem Scale (SES); Scoliosis Research Society-20 (SRS-20); Self-reported Flexilevel Scale of Shoulder Function; Spinal Function Questionnaire (Spine Score); Sporting Activity Questionnaire (Sport Score); Ste-Justine Body Image Questionnaire; Three-level version (3LY); Trauma Symptom Checklist for Children—Alternative (TSCC-A); Wechsler Intelligence Scale for Children-Revised (WISC-R); Work status question; World Health Organisation—Five Well-being index; Youth Self-Report (YSR); and Zung Self-Rating Depression Scale

In the ADS population, the following tools were used in only one study: Charlson Comorbidity Index (CCI), Fear Avoidance Beliefs Questionnaire (FABQ), Graphical Rating Scale (GRS) for worst back or leg pain, Modified Oswestry Low Back Pain Disability Questionnaire v.1.2, Modified Prolo Scale + Patient Satisfaction Index (PSI), Numeric Visual Scale (NVS), and Walking Ability.

In the Adult IS population, the following tools were used in only one study: Global perceived effect (GPE); Graphical Rating Scale (GRS) for worst back or leg pain; Modified Oswestry Low Back Pain Disability Questionnaire v.1.2; Modified Prolo Scale + Patient Satisfaction Index (PSI); Pain drawing, presence, side(s), and level(s) of radicular pain; Radicular pain scores; and the Short Form-12 Version 2 (SF-12V2).

In the ASD population, the following tools were used in only one study: Cumulative Illness Rating Scale (CIRS), Modified Japanese Orthopaedic Association (mJOA), Numeric Back Pain Scores, Numeric Leg Pain Scores, PROMIS Satisfaction with Participation in Social Roles V1.0, and the RAND-36 Questionnaire.

In the unclearly defined scoliosis population, the following tools were used in only one study: Back-pain-specific version of Hannover Functional Ability Questionnaire (HFAQ), Quadruple Numeric Pain Scale (QNPS), and the Dickson’s Standardized Questionnaire.

### 3.3. Most Common PROMs Languages

Overall, English was the most frequently used PROM language in our retrieved articles (232 studies), followed by Chinese (37), Japanese (25), Polish (22), German (18), Spanish (18), Turkish (17), Italian (16), French (14), Swedish (13), and Korean (9). A total of 22 PROM languages were encountered (Appendix C: Table A2 Alphabetical list of PROMS inventoried with record of the language translations used in the articles included). The language used to present the PROMs to participants could not be identified in 60 articles. For the SRS-22 and SRS-22r, 17 and 14 language translations were used, respectively, in patients with scoliosis treated non-operatively. Similarly, the ODI was used in 16 languages, and the SF-36 was used in 14 languages. The frequency with which each language was encountered for each tool appear in Appendix C Table A2.

### 3.4. Number of PROMS Identified per Domains

A wide group of 16 domains were assessed in patients with scoliosis treated non-operatively. Furthermore, a large variability of PROMs was used to assess the domains of Disability, Pain, Quality of life, Psychological, and Perceived appearance status (Table 4).

### 3.5. Most Commonly Targeted PROMs Domains in the Most Common Languages

When combining the populations, the outcome domains most targeted by PROMs in English articles were quality of life (246 articles), activity limitation (119), and pain (78) (likely due to the high usage of the SRS-22 versions (163) and SF-36 (54) for quality of life, the ODI (113) for activity limitation, and the SRS-22 versions and VAS scales for pain). These results are consistent with both of the next two most common PROM languages administered. For Chinese, 22 studies assessed quality of life, 11 assessed activity limitations, and 11 assessed pain. In Polish PROMs, 22 articles measured quality of life, 5 measured the activity limitation domain, and 3 focused on pain.

### 3.6. Stages of Non-Operative Care Most Often Monitored with PROMS

The description of the non-operative care studied in the included PROMs studies was often unclear (20 studies) or with care simply described as conservative, non-operative, or non-surgical or by exclusion of surgical care (52 studies). Most of the PROMS studies that were included qualified because they reported pre-operative measurements (289 studies). Bracing was the next most common treatment stage studied (48 studies), followed by observation (76) and then by exercises (48). Some adult studies also included a focus on medications or injections, but most studies including these treatments were described broadly as offering non-operative care.

## 4. Discussion

There is not yet a database of PROMs to monitor the effects of non-operative treatments and their available languages for each scoliosis sub-population. The aim of this scoping review was to create an inventory of the PROMs used for the non-operative care of patients with AIS, ASD, ADS, and adult IS. Faraj et al. 2017 had only identified PROMs for patients treated with surgery and only within the ASD population [18]. To our knowledge, no such inventory is available in the context of scoliosis populations treated non-operatively. Furthermore, this scoping review also aimed to determine the most targeted outcome domains combining the scoliosis sub-populations and for which non-operative treatment approaches PROMS were most often used. We defined non-operative care as the use of non-surgical treatment which includes bracing and/or exercises. We also included studies that provided pre-operative PROM data when the patient did not have a history of surgical intervention. Creating a list of all the current PROMs was necessary to plan an ongoing systematic review of their measurement properties which will help to identify which PROM has the best metrological properties. We will thereby be able to provide evidence-informed recommendations for consistent measurements of non-operative treatment outcomes across different languages and populations.

Based on our search, 145 PROM tools were identified. The search allowed for the identification of candidate tools for consideration in our systematic review and for a collaborative database. This list included PROMS that have been proposed more recently or used in languages other than English or targeting outcome domains which are less commonly assessed. This will ensure a thorough assessment of the most promising options.

Overall, the most frequently used PROMs were the different Scoliosis Research Society Questionnaire versions, ODI, Short Form Questionnaire versions, and Visual Analog Scales. Faraj et al.’s 2017 [18] systematic review concluded similarly that the most frequent PROMs used for surgical care of ASD patients were the SRS-22 and the ODI. The SRS-22 questionnaire addresses overall quality of life through examination of self-image, physical functioning, perceived self-appearance, mental health, and satisfaction with treatment for patients with scoliosis [21]. The ODI was developed to evaluate the activity limitations of a patient with low back pain [22]. In a 2017 Delphi consensus exercise among surgeons from Nordic countries, De Kleuver et al. recommended the SRS-22r and the EQ-5D because they contributed to measuring 10 of the 13 core outcome domains for patients with scoliosis [16]. Although the SRS-22 recommendation was supported by their finding numerous psychometric studies, this was not the case for the EQ-5D, which had yet to be studied in scoliosis participants. Furthermore, this study focused only on surgical patients and excluded patients treated non-operatively. The study may also present limited generalizability due to only focusing on the Nordic population. In our scoping review on non-operative care, the more recently proposed improvement, the SRS-22r, was used less than the SRS-22 (70 times vs. 170), and the EQ-5D was used 15 times. It is possible that the SRS-22r with its modified function domain to improve its measurement properties [21] was misreported as the SRS-22 [23] in many studies in our inventory. We recommend that in the future, authors specifically identify which SRS-22 version they are using. The EQ-5D is more complex to administer and, to date, has been used less often [24].

As expected for a search of the literature in MEDLINE, English was the most frequent language used among the studies found. Interestingly, the next more common languages were Chinese and then Japanese. There were no differences in the types of outcomes most frequently used between these languages. We identified quality of life and activity limitations as being the most frequently targeted outcome domains in all languages. Our systematic review of the measurement properties to follow will more clearly identify the gaps in the cross-cultural validation of the tools. The gaps in cross-cultural validation appear important. Only a few tools are available in multiple languages, and given that an international collaboration would likely include many European, Asian, and North American countries with research infrastructure, many more language adaptations will be needed for the most promising tools.

The AIS population was studied in 274 papers. The SRS-22, which focuses on HRQOL parameters including mental health, function, pain, and self-perceived image, was used in almost half (40.5%) of these studies. ADS was studied in 65 papers; of those, the ODI was used in over three quarters (78.5%) of the studies and focuses on the subjective degree of disability in activities of daily living. VAS, which is a subjective measure of perceived pain, was also used in (40.0%) of papers on ADS. Clearly, the preoccupations in this population differ from AIS. ASD was reported in 116 papers; of those, the ODI was used in 84.5% of the studies, whereas the SF-36 was used in 37.9% of the studies. The SF-36 focuses on eight domains of health, such as limitations in physical, social, and usual activities; pain; mental health; vitality; and general health perceptions. The domains targeted by non-operative treatments in this heterogeneous population are also slightly different but relate more closely to those targeted for ADS than AIS. Adults with idiopathic scoliosis were studied in much fewer papers (17); of those, the ODI was used in all papers (100.0%), whereas the SRS-22 was used in half (52.9%). This could illustrate that this population presents lower non-operative care needs, but those needs appear to represent a blend of what was observed in the AIS and the ADS populations.

Interpreting differences that were found suggests that AIS studies focused mostly on HRQOL using a scoliosis-specific PROM, whereas the ADS studies focused primarily on the degree of disability and pain with their condition. Interestingly, there were differences among the different adult populations in the outcomes used. The mixed ASD population used ODI for activity limitations and SF-36 for HRQOL most frequently, whereas studies in adults with IS tended to focus on pain with VAS and HRQOL with the SRS-22 while also measuring activity limitations with the ODI. Clinicians may wish for a clear recommendation of which tools to use for each language and with each population. Unfortunately, the present review only documented frequency of use of each PROM within different populations and outcome assessment domains and languages. Therefore, adopting the SRS-22r, the ODI, and VAS for pain may be recommended to allow comparisons with most prior studies while capturing the top three most studied outcomes domains. However, evidence-based or consensus-based recommendations cannot be made in the present study. Further research is needed to make recommendations on the basis of the measurement properties of the PROMs and their relevance to patients and clinicians.

The observation that research in the different scoliosis populations focuses on different outcome domains suggests it is important to clearly identify the population included in different studies. Unfortunately, an important portion of the research in adults mixes patients with different types of scoliosis. Indeed, we found more studies on the vaguely defined ASD group than in adults with IS. Given that ADS and adults with IS appear to present different concerns with more pain issues affecting ADS, studies should report results separately for these two populations. Wrong patient population was also the most important reason for exclusion from this scoping review. Many papers were excluded if they mixed patients with and without surgeries and if they mixed patients with coronal disorders with those that have purely sagittal alignment disorders.

In our study, we noted a significant barrier to classifying patient populations. For example, ASD is defined by the International Spine Study Group (ISSG) as the presence of 1 or more of the following: coronal curve Cobb angle >20 degrees, sagittal vertical axis >5 cm, pelvic tilt >25 degrees, or thoracic kyphosis >60 degrees [25,26]. This definition remains consistent across studies within the ISSG and the European Spine Study Group (ESSG) [27,28]. However, there appears to be no consensus on a clear definition of ASD across other studies. For example, a systematic review by Teles et al., 2017 defined ASD as an angular value of more than 10 degrees in the coronal plane present after spinal maturity [29]. To properly assess patients with ASD, we need to come to a consensus on a clear definition of ASD or simply eliminate this heterogeneous label and adopt more specific diagnoses. Although sagittal profile changes have been shown to influence HRQOL [30], different populations presenting with differing spinal changes should be analyzed separately until evidence shows they can be combined. Scoliosis research societies are encouraged to develop reporting guidelines for adult studies as has been accomplished for research in AIS [31].

There are limitations to this study. The search was limited to the Medline database and was conducted using only English search terms, and only reports published in English were included. If the search included similar terms in different languages and in other databases, it is possible that more articles using PROMs in other languages could have been found and included. In this scoping review, the articles were screened by a single reviewer. Surprisingly, many studies do not specify the patient population examined and/or the language used. We recommend journal reviewers insist on the authors specifying these important methodological elements. We also focused our search on PROMs and did not include outcome measures that could be performed by clinicians. Future research is planned to inventory clinician recorded impairment, disability, and participation measurements.

## 5. Conclusions

This scoping review of PROMs used for the non-operative care of AIS, adult degenerative scoliosis, ASD, and adults with idiopathic scoliosis has provided an inventory of each PROM’s name and of the languages and the populations in which they were used, as well as with what non-operative treatment type these PROMS were used to document outcomes. PROMS used to assess non-operative treatments assess a variety of outcome domains, and the most studied domains vary between patient populations. Next, it is necessary to determine which PROMs demonstrate the best measurement properties for non-operatively managed scoliosis. Our goal is to use this inventory to develop an exhaustive search as we conduct a thorough review of the measurement properties of PROMs. Such a review will inform a consensus effort to select PROMS for a collaborative database of core outcome measures to assess any patient with scoliosis undergoing non-operative management. 

## Figures and Tables

**Figure 1 children-10-00239-f001:**
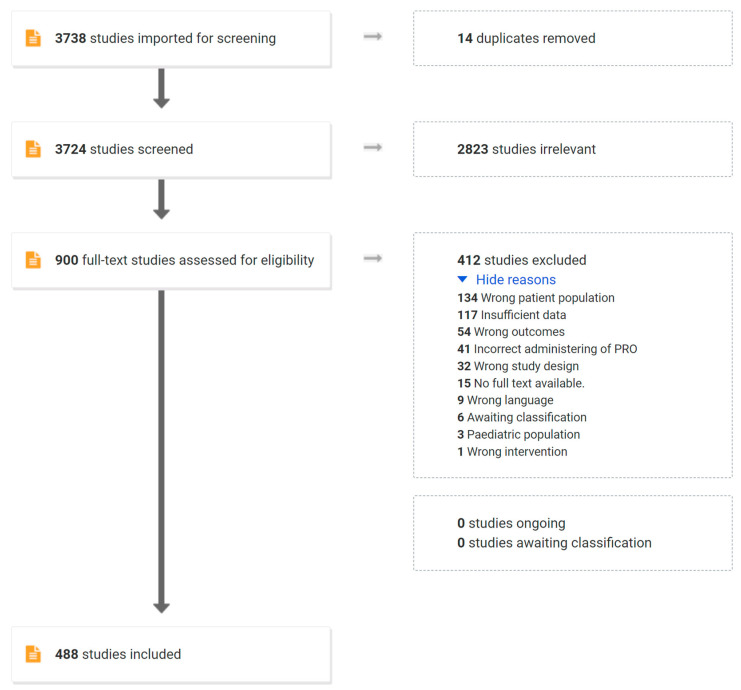
Study selection flow chart.

**Table 1 children-10-00239-t001:** Study selection criteria.

Inclusion Criteria	Exclusion Criteria
Idiopathic or degenerative scoliosis or adult spinal deformity	Other diagnoses in more than 20% of the sample
Age 10 or over	Other ages in more than 20% of the sample
Includes Patient Reported Outcomes (questionnaires that the patient answers for themself to assess their health status)	Article types on PROMs for which no quantitative data were presented.
Full-text articles published in English	Outcomes solely meant for peri- or post-operative measurements.
	Case studies, case reports, editorials, non-human studies, simulation studies, conference proceedings, abstracts, and commentaries

**Table 2 children-10-00239-t002:** Frequency of PROM used more than once (number of studies) overall and by population in the Adolescent Idiopathic Scoliosis (AIS), Adult Degenerative Scoliosis (ADS), Adult Idiopathic Scoliosis (Adult IS), Adult Spinal Deformity (ASD), and Unclear samples.

Patient Reported Outcome Measure	Overall	AIS	ADS	Adult IS	ASD	Unclear
Oswestry Disability Index (ODI)	182	21	51	17	98	13
Scoliosis Research Society-22 (SRS-22)	170	111	8	9	40	10
Short Form-36 (SF-36)	98	30	12	2	44	4
Visual Analogue Scale (VAS)	75	23	26	3	19	5
Scoliosis Research Society-22r (SRS-22r)	70	34	1	1	34	2
Scoliosis Research Society-24 (SRS-24)	27	22	0	1	2	4
Spinal Appearance Questionnaire (SAQ)	27	25	0	1	0	0
Bad Sobernheim Stress Questionnaire-Deformity (BSSQ)	24	15	0	0	0	0
Scoliosis Research Society-30 (SRS-30)	21	14	1	0	5	1
Short Form-12 (SF-12)	21	4	5	4	8	3
Numeric Rating Scale (NRS)	18	2	4	4	8	2
Bracing Questionnaire (BrQ)	15	15	0	0	0	0
European Quality of Life-5 Dimensions (EQ-5D)	15	10	2	0	2	2
Trunk Appearance Perception Scale (TAPS)	13	12	0	0	0	1
Core Outcome Measures Index (COMI)	11	1	4	2	6	0
Scoliosis Research Society (SRS)	10	1	3	1	5	3
Scoliosis Research Society-23 (SRS-23)	9	7	0	0	1	1
Body Image Disturbance Questionnaire (BIDQ)	8	5	0	0	0	2
Roland-Morris Disability Questionnaire (RMDQ)	7	3	4	1	0	0
Global Rating of Change Scale (GROC)	5	3	0	0	1	1
Pediatric Quality of Life (PedsQL)	5	5	0	0	0	0
Quality of Life Profile for Spine Deformities (QLPSD)	5	4	0	0	0	1
Walter Reed Visual Assessment Scale (WRVAS)	5	5	0	0	0	0
World Human Organization QOL Questionnaire (WHOQOL-BREF)	5	4	0	0	0	1
Lumbar Stiffness Disability Index (LSDI)	4	1	1	0	2	0
Pediatric Outcomes Data Collection Instrument (PODCI)	4	3	0	0	0	1
PROsetta Stone crosswalk tables for Patient-Reported Outcomes Measurement Information System (PROMIS) Pain Interference (PI)	4	2	0	0	2	0
PROsetta Stone crosswalk tables for Patient-Reported Outcomes Measurement Information System (PROMIS) Physical Function (PF)	4	1	0	0	3	0
Spielberger State-Trait Anxiety Inventory	4	4	0	0	0	0
Child Health Questionnaire—Child Form 87	3	2	0	0	0	1
EuroQol—Visual Analogue Score (EQ-VAS)	3	1	0	0	1	1
Functional Rating Index (FRI)	3	2	0	0	0	1
Japanese Orthopaedic Association (JOA)	3	0	3	0	0	0
Japanese Orthopaedic Association Back Pain Evaluation Questionnaire (JOABPEQ)	3	2	0	0	1	0
Neck disability index (NDI)	3	1	0	0	1	1
Pain catastrophizing scale	3	2	0	1	0	0
Patient Health Questionnaire (PHQ-9)	3	2	0	0	0	1
Revised Oswestry Back Pain Disability Questionnaire (OSW)	3	2	0	0	0	1
Scoliosis Research Society-29 (SRS-29)	3	0	0	0	2	1
Scoliosis Research Society-7 (SRS-7)	3	2	0	0	1	0
Short Form of McGill Pain Questionnaire (MPQ-SF)	3	2	0	0	0	1
16PF Adolescent Personality Questionnaire (16PF APQ)	2	2	0	0	0	0
Children’s Depression Index (CDI)	2	2	0	0	0	0
General Function Score (GFS)	2	1	0	0	0	1
Global Back Disability Question	2	1	0	0	0	1
Mini Mental State Examination (MMSE)	2	0	0	0	2	0
Oswestry Low Back Pain Disability Questionnaire (ODQ)	2	2	0	0	0	0
Patient-Specific Functional Scale (PSFS)	2	1	0	0	0	1
Pediatric Patient-Reported Outcome Measurement System (PROMIS) Short Forms for fatigue, depression, anxiety, pain interference, and mobility	2	2	0	0	0	0
Pediatric Quality of Life 4.0 (PedsQol4.0)	2	2	0	0	0	0
Piers–Harris Self Concept	2	2	0	0	0	0
Scoliosis Research Society-Quality of Life (SRS-QOL)	2	0	0	0	0	2
Short Form-36 Version 2 (SF-36V2)	2	0	1	0	2	0
Short Form-6 Dimension (SF-6D)	2	0	1	0	1	0
Strengths and Difficulties Questionnaire (SDQ)	2	2	0	0	0	0
Tampa Scale for Kinesiophobia (TSK)	2	0	1	1	0	0

**Table 3 children-10-00239-t003:** Frequency of the 10 most frequently used PROMs, which are expressed as a percentage of the number of articles they appeared in for each population.

Rank	Overall (488 Studies)	AIS (274 Studies)	ADS (65 Studies)	Adult IS (17 Studies)	ASD (116 Studies)
1	ODI (37.3%)	SRS-22 (40.5%)	ODI (78.5%)	ODI (100.0%)	ODI (84.5%)
2	SRS-22 (34.8%)	SRS-22r (12.4%)	VAS (40.0%)	SRS-22 (52.9%)	SF-36 (37.9%)
3	SF-36 (20.1%)	SF-36 (10.9%)	SF-36 (18.5%)	SF-12 (23.5%)	SRS-22 (34.5%)
4	VAS (15.4%)	SAQ (9.1%)	SRS-22 (12.3%)	NRS (23.5%)	SRS-22r (29.3%)
5	SRS-22r (14.3%)	VAS (8.4%)	SF-12 (7.7%)	VAS (17.6%)	VAS (16.4%)
6	SRS-24 (5.5%)	SRS-24 (8.0%)	RMDQ (6.2%)	SF-36 (11.8%)	SF-12 (6.9%)
7	SAQ (5.5%)	ODI (7.7%)	NRS (6.2%)	COMI (11.8%)	NRS (6.9%)
8	BSSQ (4.9%)	BSSQ (5.5%)	COMI (6.2%)	*13 other*	COMI (5.2%)
9	SRS-30 (4.3%)	BrQ (5.5%)	SRS (4.6%)	*PROMs with*	SRS (4.3%)
10	SF-12 (4.3%)	SRS-30 (4.3%)	JOA (4.6%)	*1 study each*	SRS-30 (4.3%)

Abbreviations: 22r—22 revised, BrQ = Brace questionnaire, BSSQ = Bad Sobernheim Stress Questionnaire, COMI = Core Outcome Measure Index, JOA = Japanese Orthopedic Association, ODI = Oswestry disability index, NRS = numeric rating scale, RMDQ = Roland Morris Disability Questionnaire, SAQ = spinal appearance questionnaire, SF = short form, SRS = Scoliosis Research Society, VAS = visual analog scale.

**Table 4 children-10-00239-t004:** Inventory of the assessment domains targeted in the included articles with the number of PROMs inventoried for each domain.

Assessment Domains	Number of Inventoried PROMs
Activity limitation	30
Pain	25
Quality of Life	33
Psychological	25
Perceived appearance	10
Perceived change	2
Physical activity	3
Satisfaction	2
Fear avoidance	2
Predict adherence to brace-wear	2
Multi-domains	2
Comorbidities	3
Sleep	3
Fatigue	1
Health status	1
Dyspnea	1
Intelligence	1

## Data Availability

Not applicable.

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
