# Peer review of "Inventory of Patient-Reported Outcome Measures Used in the Non-Operative Care of Scoliosis: A Scoping Review"

_children, 2023, doi:10.3390/children10020239_

Round 1
Reviewer 1 Report
Manuscript : Inventory of Patient-reported Outcome Measures Used in the Non-operative Care of Scoliosis: A Scoping Review.
The objective of this scoping review was to create an inventory of the patient- and proxy-reported outcome measures (PROMs) used in monitoring the effects of non-operative care management of adolescents and adults with scoliosis. Evaluation of the effects of therapy is a very important aspect of medical care for patients with scoliosis. Detailed assessment and appropriately planned treatment often reduce the risk of scoliosis progression, accompanying disorders and expensive surgery.
The introduction contains all relevant information related to the topic. The negative impact of scoliosis on everyday functioning was demonstrated in detail and the importance of the analysis of various PROMs used in the assessment of patients with scoliosis treated conservatively was justified.
Search strategy, screening strategy with inclusion/exclusion criteria and data processing were precisely described.
A total of 145 PROMs were identified in 22 different languages. These results are an advantage of the review. The additional value is the list of PROMs used in various populations of people with scoliosis. The authors emphasized that only a few tools are available in multiple languages ​​to monitor the effects of non-operative treatment of scoliosis.
The results of this scoping review, including an inventory of PROMs, can be used to develop recommendations for measurements of non-operative treatment effects in various populations of people with scoliosis.
In general, the manuscript is relevant for the field and eligible for publication with some corrections:
Introduction
Lines 99-100 and 109-111 – in these lines, the information that this scoping review will provide a basis for the systematic review is repeated twice. It is not necessary to repeat this information.
Line 124 – What is the meaning of the sentence “To limit the yield to the most relevant references”. Is the construction of this sentence correct?
Results
Figure 1 - What does it mean “Wrong language”? (column -> 412 studies excluded / reasons). Explain please.
Line 172 – what does it mean the term “the Unclear category”? Explain please.
Table 3. I don't understand the description of most frequently used PROMs in the column “Adults”, Rank 8 (13 other), Rank 9 (PROMs with), Rank 10 (1 study each) in the Table 3. (Title: Frequency of the 10 most frequently used PROMs, expressed as a percentage of the number of articles they appeared in for each population). Could you clarify, please?
Line 201-202 – “The PROMs language used in 60 articles could not be identified”. Have these articles been excluded? The study selection flow chart (Figure 1) does not contain this information.
Line 216-220 – “…the ODI (113) for function and the SRS-22 versions and VAS scales for pain). These results are consistent with both next two most common PROM languages administered. For Chinese, 22 studies assessed quality of life, 11 function and 11 pain. In Polish PROMs, 22 articles measured quality of life and 5 function domain and 3 focused on pain. …”. What is the meaning of the word “function” in the manuscript? Do you mean body functions or activities according to International Classification of Functioning, Disability and Health (ICF, WHO)?
According to ICF: Body functions - The physiological functions of body systems (including psychological functions); Body structures - Anatomical parts of the body such as organs, limbs and their components: Impairments - Problems in body function and structure such as significant deviation or loss; Activity - The execution of a task or action by an individual; Participation - Involvement in a life situation (e.g. personal care - washing and dressing, lifting, walking, sitting are activities of daily living). Could you explain please?
Discussion
Line 258-259 – “ The ODI was developed to evaluate the function of a patient with low back pain[22]”. Function or activities of daily living (according to ICF)?
Line 266-267 - “We identified quality of life and function as being the most frequently targeted outcome domains in all languages”. What is the meaning of “function”?
Line 284-286 – “The AIS population was studied in 274 papers. The SRS-22, which focuses on HRQOL including mental health, function, pain, and self-perceived image, was used in almost half (40.5%) of these studies”. What “function” means in this sentence?
Author Response
See fille uploaded

Reviewer 2 Report
It is an interesting paper that reviews the most used PROMs in the conservative treatment of AIS and adult degenerative scoliosis. The amount of literature reviewed (n=488) and PROMs identified (145) is clearly the main strength of this research. It´s main contribution is to have identified ODI, SRS-22 and SF-36 as the most used PROMs in an overall 5 different populations and 22 languages
Search strategy and study selection criteria were clearly exposed in the methods section. My congratulations to the reviewer that extract all data from the 488 studies… such a hard work!
I highly appreciate the hard work done in this paper to identify the most used PROMs and domains in studies of conservative treatment of these types of idiopathic scoliosis. And I also noticed that a further work will be done in the direction of elaboration of a consensus of PROM selection in non-operative patients.
But I´d like to extract from all the information collected some clinical applications about which PROM is suspected to be better in AIS, ADS, ASD or adult IS under the author´s opinion and expertise. Probably the presence/absence of pain, i.e., may led to different elections (which has been evidenced in the review), but other criteria could be possible. With the results presented I concluded that ODI, SRS-22 and SF-36 could be the most useful, probably SRS-22 in AIS and ODI in adults, but do the authors agree with this conclusion? Do they have their own one?
- Specific comments
In lines 37-39 brace and surgery, and their costs, are exposed. I miss something about PSSE, which is both the adjuvant treatment of bracing and also have a great economic cost to families.
Table 2, which reports PROMs identified during the literature exhaustive revision, show that 89 of the 145 PROMs are present only in 1 overall study. I would suggest the authors to sum up this information to reduce table size (currently almost 5 pages of 29). Probably by writing this information in the text instead of using table rows that are not too informative. If the scope of the review is to identify the most used PROMs in scoliosis studies, the ones reported in one or even 2 studies (15 of 145) should not be clinically relevant. My conclusion from the table is that the 71% of PROMs are not very useful (only reported once or twice in the literature) and it could be drawn with a reduced table that facilitates reading and the capture of the main message.
Table 3 is, under my opinion a high-valuable result for clinicians, as they can observe how, for example, SRS-22 is the most used PROM in AIS while ODI is in 7th position. Oppositely, in the rest forms of scoliosis, ODI is the most PROM reported while SRS-22 is less used. I´d like to thank the authors for this contribution!
Author Response
See our response in the uploaded file.

Reviewer 3 Report
Dear Authors,
This is a well-designed and very useful study. Many aspects of the topic had been exposed in an appropriate way. Statistic strategy fits well the purposes of the study. Literature knowledge is good. The conclusion is in line with the results shown in the paper. Language is good for a scientific paper, but can be improved.
Target population is a bit wide and not very well explained in the methods. Is there the possibility to be more precise in the methods?
Study design: scoping review.
Aim: create an inventory of the PROM tools used in monitoring the effects of non-operative care management of AIS and adults with structural spine disorders.
Target population: patients with AIS and adults with structural spine disorders.
Collection: all papers expect “ case studies, case reports, editorials, non-human studies, simulation studies, conference proceedings, abstracts, and commentaries “.
Materials and Methods
The population target, the intervention, the collection and outcomes has been exposed. Nevertheless, it is not very clear population target. Therefore
Table 1: where exclusion is “Other diagnoses in more than 20% of the sample” is not completely clear to me.
Line 163: what is irrelevant study design? This: Case studies, case reports, editorials, non-human studies, simulation studies, conference proceedings, abstracts, and commentaries?
Data extraction and synthesis: these aspects are defined and explained well in the methods sections.
Results
The results are exposed well and are clear.
163: why the irrelevant studies were eliminated in the full-text screening?
164: please explain better this last sentence. It is not clear to me what it was done and the meaning of this. Did you take a paper of reference and checked the ref. list to see if all papers were included in your search?
Table 2: the last column is written unclear? It means that it is unclear the disease? In this case should not they been excluded?
Discussion
The literature is well organized and the choice of the papers is large enough.
315: this was not an exclusion criteria right? Why were these paper excluded? In the exclusion was a reason a “solely meant for peri- or post-operative measurements”.
Conclusion
Conclusion is in line with the findings shown in the results. The author could answer to their research question.
Author Response
See our response document file uploaded below.

Author Response
See attached our response to reviewer 4 file.
